# Seaweed and Seaweed Bioactives for Mitigation of Enteric Methane: Challenges and Opportunities

**DOI:** 10.3390/ani10122432

**Published:** 2020-12-18

**Authors:** D. Wade Abbott, Inga Marie Aasen, Karen A. Beauchemin, Fredrik Grondahl, Robert Gruninger, Maria Hayes, Sharon Huws, David A. Kenny, Sophie J. Krizsan, Stuart F. Kirwan, Vibeke Lind, Ulrich Meyer, Mohammad Ramin, Katerina Theodoridou, Dirk von Soosten, Pamela J. Walsh, Sinéad Waters, Xiaohui Xing

**Affiliations:** 1Lethbridge Research and Development Centre, Agriculture and Agri-Food Canada, 5403-1 Avenue South, Lethbridge, AB T1J 4B1, Canada; wade.abbott@canada.ca (D.W.A.); karen.beauchemin@canada.ca (K.A.B.); rob.gruninger@gmail.com (R.G.); mike.xing@uleth.ca (X.X.); 2Department of Biotechnology and Nanomedicine, SINTEF Industry, 7465 Trondheim, Norway; Inga.M.Aasen@sintef.no; 3Department of Sustainable Development, Environmental Science and Engineering, KTH Royal Institute of Technology, 114 28 Stockholm, Sweden; fredrik.grondahl@abe.kth.se; 4Food BioSciences Department, Teagasc Food Research Centre, Ashtown, D15 KN3K Dublin 15, Ireland; 5Queens University Belfast (QUB), Belfast, BT7 1NN Co., Antrim, Ireland; s.huws@qub.ac.uk (S.H.); K.Theodoridou@qub.ac.uk (K.T.); pamela.walsh@qub.ac.uk (P.J.W.); 6Animal Bioscience Research Centre, Grange, Dunsany, C15 PW93 Co., Meath, Ireland; David.Kenny@teagasc.ie (D.A.K.); Stuart.Kirwan@teagasc.ie (S.F.K.); sinead.waters@teagasc.ie (S.W.); 7Department of Agricultural Research for Northern Sweden, Swedish University of Agricultural Sciences, SE-901 83 Umeå, Sweden; sophie.krizsan@slu.se (S.J.K.); mohammad.ramin@slu.se (M.R.); 8Norwegian Institute of Bioeconomy Research (NIBIO), Post Box 115, 1431 Ås, Norway; vibeke.lind@nibio.no; 9Friedrich-Loeffler-Institut (FLI), Bundesforschungsinstitut für Tiergesundheit, Federal Research Institute for Animal Health, 38116 Braunschweig, Germany; Ulrich.Meyer@fli.de (U.M.); Dirk.von_Soosten@fli.de (D.v.S.)

**Keywords:** methane emissions, rumen, ruminants, seaweeds, bioactive components, bromoform, bacteriocins, peptides, carbohydrates, lipids, saponins, phlorotannins, alkaloids, animal studies, RUSITEC

## Abstract

**Simple Summary:**

The need to become more efficient in agriculture and the food industry exists parallel to the challenge of climate change. Meat and dairy production is the target of much scrutiny due to methane (CH_4_) emissions and global warming. On the other hand, it should be noted that two-thirds of the world’s agricultural land consists of pastures and permanent grasslands and is used for livestock grazing. This land is predominantly unsuitable for arable purposes but facilitates the production of high-quality human-edible protein in the form of ruminant animal-derived meat and milk. This makes a significant contribution to feeding the world’s population. There is a need to reduce CH_4_ emissions, however, and several approaches are being researched currently. Seaweeds are diverse plants containing bioactives that differ from their terrestrial counterparts and they are increasingly under investigation as a feed supplement for the mitigation of enteric CH_4_. Seaweeds are rich in bioactives including proteins, carbohydrates and to a lesser extent lipids, saponins, alkaloids and peptides. These bioactives could also play a role as feed ingredients to reduce enteric CH_4_. This review collates information on seaweeds and seaweed bioactives and their potential to impact on enteric CH_4_ emissions.

**Abstract:**

Seaweeds contain a myriad of nutrients and bioactives including proteins, carbohydrates and to a lesser extent lipids as well as small molecules including peptides, saponins, alkaloids and pigments. The bioactive bromoform found in the red seaweed *Asparagopsis taxiformis* has been identified as an agent that can reduce enteric CH_4_ production from livestock significantly. However, sustainable supply of this seaweed is a problem and there are some concerns over its sustainable production and potential negative environmental impacts on the ozone layer and the health impacts of bromoform. This review collates information on seaweeds and seaweed bioactives and the documented impact on CH_4_ emissions in vitro and in vivo as well as associated environmental, economic and health impacts.

## 1. Introduction

Livestock supply chains emit 3.1 gigatonnes CO_2_-eq of CH_4_ per annum, or 44 percent of anthropogenic CH_4_ emissions [1]. On the other hand, it should be noted that two-thirds of the world’s agricultural land consist of pastures and permanent grasslands and is used for livestock grazing and this contributes significantly to meeting the nutritional requirements of an ever growing global human population [2,3,4]. The rumen is a complex ecosystem composed of bacteria, fungi, protozoa, methanogens and bacteriophages, all of which contribute to dietary energy harvesting and resultant nutrient supply to the host. CH_4_ is a by-product of this fermentative process due to released hydrogen being utilised by the methanogens to produce CH_4_. Changes in ruminant diets can affect the rumen microbiome and their digestive capacity that in turn affects CH_4_ emissions. Numerous reports have shown that the composition of the ruminant diet can cause changes in rumen microbial diversity and alter methanogenic activity and gas production [5,6]. CH_4_ release by rumen methanogens has also been reported to contribute to dietary energy losses of between 2 and 12% [7]. Although, rumen methanogens negatively impact feeding efficiencies of livestock through digestion energy losses, mitigation measures to reduce CH_4_ emissions through diet do not necessarily translate to reduced CH_4_ emissions in all instances. A previous review of the literature by van Gastelen et al. [8] showed the effectiveness of forage-related CH_4_ mitigation strategies, such as feeding highly digestible grass and replacing different forage types with corn silage, differs across ruminant types. These strategies were found to be most effective in dairy cattle, with fewer benefits observed in beef cattle and sheep.

Several seaweeds have been identified to date with potential to reduce CH_4_ emissions from ruminants including *Asparagopsis taxiformis*, *Alaria esculenta*, *Ascophyllum nodosum* and *Chondrus crispus* [9,10,11,12]. CH_4_ reduction is largely attributed to the compound bromoform [12], which is found in several seaweed species especially red seaweeds like *Asparagopsis* spp. Bromoform is known to inhibit the CH_4_ biosynthetic pathway within methanogens [12]. However, other compounds including carbohydrates, lipids, peptides and phlorotannins have also been identified as having the potential to inhibit methanogens including Archaea [13] and potentially could reduce CH_4_ emissions and improve animal health and production through their activities.

Recently, researchers concluded that commercial production of the red seaweed *A. taxiformis* could create new economies due to the fact that addition of small quantities of this seaweed in the diet of ruminant animals reduced CH_4_ emissions by up to 98% when included at 0.2% of dry matter intake of steer diets [14]. This result was attributed to bromoform, which acts by inhibiting methanogens, while not affecting other bacteria [15]. However, bromoform is a known carcinogen and has also been reported to impact negatively on the ozone layer. Furthermore, it is not known if inhibition of methanogens by bromoform is temporary and furthermore, it is not known how long the CH_4_ reductions last. Moreover, *A. taxiformis* is native to South Australia but is an invasive species in the Northern hemisphere and it is currently not cultivated in large quantities in the Northern hemisphere. The aim of this review paper is to highlight the potential of other seaweeds and their components as feed ingredients for the reduction of enteric CH_4_.

## 2. Influence of the Rumen Microbiome on CH_4_ Emissions

Ruminant livestock production is dependent on the anaerobic microbial ecosystem residing in the rumen to ferment and convert human indigestible plant matter into high-quality dairy and meat products for human consumption. Members of the rumen microbiome including bacteria, archaea, protozoa, bacteriophage and fungi, have co-evolved and have a symbiotic relationship with their host to allow ruminants the ability to ferment complex plant matter [6,16,17]. However, CH_4_ is produced as a metabolic end product of enteric fermentation by ruminant methanogens [18]. While CH_4_ has a shorter half-life than carbon dioxide it is 28 times more potent in terms of global warming potential [19].

Research on the methanogenic potential of the rumen has attracted great attention in the last decade, due to the impact methanogenesis has on livestock performance and the environment [6,20]. As the sole producers of CH_4_, a plausible hypothesis would consider an increased abundance of methanogens within the rumen to be associated with a greater output of CH_4_ emissions by the animal. However, it would seem that the composition rather than size of the methanogen community in the rumen is more closely associated to CH_4_ production [18]. For example, in cattle, no differences were noted in the overall relative abundance of Archaea between high and low CH_4_-emitting dairy cows [21]. However, the same authors reported an increased relative abundance of *Methanobrevibacter gottschalkii* and *Methanobrevibacter ruminantium* was associated with high and low CH_4_-emitting animals, respectively. Similarly in sheep, specific members of the methanogen community, rather than overall abundance of Archaea, were found to be associated with CH_4_ output [22,23].

The genus *Methanobrevibacter* is the most dominant member of the rumen archaeal community [18]. The variation in the association of abundance of members of this genus, in terms of correlation to CH_4_ production, could be as a result of differences in the expression of the different forms of methyl-coenzyme M reductase (mcr) [18] which catalyses the rate limiting step of methanogenesis (Figure 1).

The Methanobrevibacter clades can be segmented into two subgroups, the Methanobrevibacter smithii-gottschalkii-millerae-thaurei (SGMT) (*M. smithii*, *M. millerae*, *M. thaueri* and *M. gottschalkii*) and *M. ruminantinum-olleyae* (RO) clade (*M. ruminantium* and *M. olleyae*) with the SGMT clade capable of synthesising both mcrI and mcrII and the RO subgroup possessing only mcrI [18,24]. Expression of both mcrI and mcrII is regulated by H_2_ availability in the rumen, with mcrI and mcrII expression occurring in the presence of low and high concentrations of ruminal H_2_ [25]. As a result, a greater presence of SGMT methanogens in the rumen could be evidence of increased abundance of rumen microbes that synthesise H_2_ [21]. Similarly, as mcrI is produced in a low H_2_ environment, a greater presence of RO methanogens may be suggestive of a rumen microbiome harbouring a lower abundance of H_2_ producing and/or greater abundance of H_2_ utilising microbes.

Similarly, the abundance of specific bacteria, particularly those that produce H_2_ are associated with CH_4_ production [18]. In sheep, three different bacterial communities (ruminotypes Q, S and H) were identified associated with CH_4_ emissions [22]. Ruminotype Q and S were correlated with low emitting sheep and harboured a greater abundance of bacterial communities associated with producing propionate and a combination of lactate and succinate respectively. In contrast, ruminotype H found in high-level CH_4_-emitting animals was characterised by a higher abundance of H_2_ producers. Key bacterial members of Ruminotype Q and S, such as *Sharpea* spp. and *Kandleria* spp., are associated with low or no H_2_ production [22] thus leading to less H_2_ being available for methanogenesis. As such, it is hypothesised that the occurrence of these bacteria in the rumen lowers the availability of H_2_ leading to a decrease in the availability of the substrate for methanogenesis [22,26]. Similarly, the production of propionate by other bacterial groups consumes ruminal H_2_, thus reducing the supply of methanogenesis substrates resulting in lower production of CH_4_ [27]. Succinate and lactate are also precursors of propionate production [28]. Wallace et al. [29] reported a four-fold reduction in the abundance of the succinate producing bacteria family Succinivibrionaceae in high compared to low CH_4_-emitting steers. Similarly, a higher abundances of bacterial groups associated with the production of propionate precursors in low CH_4_-emitting animals has been noted elsewhere [26,30]. Therefore, strategies that increase the abundance of the RO clade in the rumen may be beneficial in terms of lowering CH_4_ output.

Interrelationships between rumen fungal populations and CH_4_ emissions are also important. Clear differences in the diversity of fungal communities of high and low CH_4_-emitting animals have not been identified to date while studies in dairy cattle have failed to define clear correlations between the fungal populations present in rumen samples and CH_4_ emissions [22]. In the study of Cunha et al. [31], 73.19% of the fungal samples were identified as unclassified and therefore poor identification could contribute toward the lack of reported correlations while it was also acknowledged that sampling method could impact identification, as anaerobic fungi are more commonly found attached to feed particles [31]. Work in anaerobic digesters inoculated with fungi harvested from rumen fistulated cattle, showed a positive correlation between fungal numbers and CH_4_ generation [32] possibly indicating the concentration of the fungal population as a whole to be associated with CH_4_ emissions. Protozoa in the rumen also affect the bacteria, fungi and archaea present. Rumen protozoa display selective predation and rumen methanogenic archaea also form close associations with the rumen protozoa, particularly the order Vestibuliferida. Ciliate protozoa possess hydrogenosomes, which compartmentalise the terminal reactions of energy metabolism, resulting in hydrogen release, which allows methanogens to utilise hydrogen to form CH_4_ more effectively [33].

Many studies have shown correlations between the host genotype and CH_4_ emissions, likely linked to a degree of host control on the rumen microbiome [34,35,36]. However, a recent publication using a Bayesian approach estimated that host genetics and microbiota explained 24% and 7%, respectively, of variation in host CH_4_ levels [37]. Furthermore, a paper by Wallace et al. [29] identified a heritable subset of the core rumen microbiome that dictates dairy cow productivity and emissions. Many studies have also illustrated the effects of diet on ruminant CH_4_ emissions. For example, lipid [38,39], condensed tannin [40] and essential oil [41] supplementation of the ruminant diet have all resulted in lower CH_4_ emissions. Nonetheless the proportional contribution of diet, host genotype and management practices remains unclear. There is considerable variability in research outcomes published, largely a consequence of our inability to measure CH_4_ emissions in a high-throughput manner that allow the use of large numbers of animals, that is required to increase the power of experiments. At present dietary interventions offer the quickest potential mechanism for lowering the environmental impact of ruminant livestock production.

## 3. Overview of In Vitro Studies Which Used Seaweeds for Feed and Impact on CH_4_ Emissions

Research on supplementing ruminant diets with seaweeds to reduce CH_4_ emissions is still at an early stage, although there is an increasing body of literature examining potential mitigation effects of seaweeds using in vitro techniques. Seaweeds have had a longstanding history of use as animal feeds, particularly in Ireland, Scotland and the Nordic and Scandinavian countries including Iceland and Norway [42] and in Asia and other areas with proximity to the sea. However, in the early 20th century, their use fell out of favour as a potential livestock feed when scientists reported that they reduced animal growth due to poor digestibility of complex carbohydrates. However, those studies tended to use high inclusion rates of up to 50% seaweeds in the diet. More than 21 seaweeds have been shown to reduce CH_4_ emissions in vitro as summarised in Table 1, while others have been shown to have no mitigation effect (Appendix A). For example, seven seaweed species—namely the brown seaweeds *Alaria esculenta*, *Laminaria digitata*, *and Pelvetia canaliculata;* the red seaweeds *Mastocarpus stellatus*, *Palmaria palmata*, *and Porphyra* spp.; and the green seaweed *Acrosiphonia* spp.—collected north of the Arctic circle were assessed for nutrients and total polyphenol content, gas production kinetics and in vitro rumen fermentation [43]. The experiment was followed up by an in vivo study in sheep fed with *Porphyra* spp. at an inclusion level of 10% of dry matter (DM). Compared with diets including white clover silage or soybean meal, inclusion of *Porphyra* spp. did not change the enteric CH_4_ emissions [44]. The greatest CH_4_ mitigation potential was observed for the red seaweed *A. taxiformis* with almost complete inhibition in vitro with inclusion levels up to 16.7% of the organic matter (OM). *A. taxiformis* was highly effective in decreasing the production of CH_4_ with a reduction of 99% at doses as low as 2% OM [45,46,47,48]. Machado et al. [49] identified that bromoform was the main bioactive in *A. taxiformis* that promotes anti-methanogenic activity in in vitro studies. Other seaweeds that have been shown to have notable mitigation potential (>50% decrease) in vitro include *Cladophora patentiramea* (green seaweed) [49], *Cytoseira trinodis* (brown seaweed) [50], *Dictyota bartayresii* (brown seaweed) [48], *Gigartina* spp. (red seaweed) [51], *Padina australis* (brown seaweed) [49]) and *Ulva* spp. (green seaweed) [49] although these initial results need to be confirmed in further in vivo studies. The decrease in CH_4_ production observed with some seaweeds in vitro was accompanied by a decrease in total gas production, total volatile fatty acid production, or substrate degradability, which suggests that performance may be compromised when feeding these seaweeds to animals. There is need for further in vivo studies to confirm the potential effects of the seaweeds identified in in vitro screening studies and to measure CH_4_ output in terms of kilograms of product (milk, meat) produced from the ruminant.

### 3.1. Brown Seaweeds

Brown macroalgae are found in the deep and colder waters of the Northern hemisphere and are reported to contain over 1100 secondary metabolites and are the only algae to contain the polyphenols phlorotannins (PTs). PTs have a broad spectrum of antimicrobial activities and act particularly on the rumen cellulolytic bacterium *Fibrobacter succinogenes*. In a recent study, Ramin et al. [9] tested the protein fractions extracted from two brown species *S. latissima* and *A. esculenta* on in vitro utilisable crude protein (uCP). They showed that replacing grass silage with different levels of seaweed protein fractions increased uCP. In the same study it was found that increased levels of *A. esculenta* fractions reduced CH_4_ production in vitro. One main reason for reductions in CH_4_ production could be due to the fact that the digestibility was decreased by increased levels of *A. esculenta* in the in vitro system. It is also possible that lower digestibility of the *A. esculenta* fractions results from a higher content of tannins (polyphenols), which affects digestibility and CH_4_ production [56]. PTs constitute a heterogeneous group of molecules with variable structure and degrees of polymerisation [57], constituting up to 90% of the phenols present in brown seaweeds [58]. These compounds are analogous to condensed tannins (CT), which are found in terrestrial plants such as chestnuts and willow [59]. Whilst terrestrial CT are derived from the polymerisation of flavan derivatives, phloroglucinol (1-, 3-, 5-trihydroxybenzene) is the basic repeating unit of PTs [60]. Brown seaweeds have several characteristics that distinguishes them from red and green seaweeds. Whilst brown seaweeds are typically low in crude protein (CP) (40–180 g/kg dry matter (DM); Table 2), their content of organic minerals and an array of biologically active compounds including polysaccharides, proteins, lipids and polyphenols make them a potentially valuable, high-functioning animal feed [60].

### 3.2. Green Seaweeds

Green seaweeds have the least variety of secondary metabolites, with fewer than 300 compounds found [51]. They are usually found in rockpools of the littoral fringe or upper eulittoral zone and are subject to widely fluctuating temperatures and salinity. Green species found here include members of the genus Ulva, along with *Cladophora* spp. *U. lactuca* is a green macro alga involved in green tides observed worldwide. *Ulva* spp. blooms occur mainly in shallow waters and the decomposition of this alga can produce dangerous vapors. *U. lactuca* is a species usually resembling lettuce and contains commercially valuable components including Ulvans. Green algae belonging to species of Ulva and Entermorpha are a known source of polysaccharides with innovative structure and functional properties. *Ulva* spp. carbohydrates can also be a carbon source for microbial production of biomaterials and building blocks to produce a range of chemicals and intermediates, such as organic acids, alcohols and biomaterials. *Ulva* spp. contains phenolics, chlorophyll and carotenoids, which can be regarded as active free-radical scavengers. The green algae Chaetomorpha (Chlorophyte, Cladophorales) are characterised by unbranched heavy filaments and contains about 70 species. They are mostly rich in bioactive compounds, and are used as dietary supplements. Some have shown cytotoxicity against cancer cell lines perviously. Ethanol extracts of *Chaetomorpha* spp. possessed higher antioxidant activity compared to aqueous extracts [67]. These algae are common on all seashores and can be produced in nutrient-enriched waters.

### 3.3. Red Seaweeds

Red seaweeds are rich in gelling biopolymers such as carrageenan and agars which are also known as galactans. They also are a rich source of a number of bioactive compounds with nutritional, functional or biological features. Several bioactive peptides with heart health benefits have been reported from red seaweeds such as *Palmaria palmata* (Dulse) and *Porphyra* spp. [68,69]. The biological activities of agar oligosaccharides include antimicrobial, antiviral, prebiotic, anti-tumour, immomodulatory and anti-inflammatory as well as antioxidant activities. Red seaweeds are rich also in halogenated low molecular weight compounds, in particular brominated and chlorinated haloforms. These compounds are known to inhibit CH_4_ production (bromoform) and also to have strong antimicrobial properties and inhibit a wide range of microorganisms, including Gram-positive and Gram-negative bacteria and fungi as well as protozoa. *Chondrus crispus* (Irish moss) is a small purplish-red seaweed (up to 22 cm long) found on rocky shores and in pools. The fronds grow dichotomously from a narrow, unbranched stipe and are flat and wide with rounded tips. This seaweed is highly variable in appearance depending on the level of wave exposure of the shore and has a tendency to turn green in strong sunlight. It is best known as a source of carageenan but also produces a di-peptide citrulline-arginine [70].

### 3.4. Seaweeds and Known Impacts on the Rumen Microbiome

Thirty five years ago Colin Orpin first identified that the microbial community in the rumens of the North Ronaldsay breed of sheep, which consume almost entirely a seaweed diet on the remote island of North Ronaldsay in the Orkney Islands, Scotland, was different from pasture-fed animals [71]. The consumption of seaweed products has also been found to lead to lateral transfer of carbohydrate active enzymes from marine bacteria to human gut microbial symbionts [72]. These studies point to the potential value of including seaweed and/or seaweed extracts to manipulate the composition of the rumen microbiome. A screen of the effect of 20 different fresh and marine macroalgae on in vitro rumen fermentation found the impact of the different seaweeds was highly variable and identified the brown seaweed *Dictyota* spp. and red seaweed *A.taxiformis* as causing a significant reduction in CH_4_ production [46]. In vitro studies examining the effect of both *A. taxiformis* and the secondary metabolite bromoform found that both treatments had targeted effects and reduced the abundance of methanogenic archaea in rumen fluid [53]. Analysis of the effects of *A. taxiformis* on rumen fermentation and the composition of the microbiome was recently examined using a semi-continuous in vitro rumen system [14]. Inclusion of 5% *A. taxiformis* in a dairy ration resulted in a 95% reduction in CH_4_ emissions with no negative impacts on rumen fermentation. Only small shifts in the microbial community were observed with the abundance of methanogenic archaea remaining similar between the control and *A. taxiformis* treatments. This study highlights that large-scale changes in the rumen microbial community are not a prerequisite for altering the function of the rumen microbiome. Much of the focus of the studies involving supplementation of seaweed into the diet of ruminants has focused on the potential anti-methanogenic effects. However, there is also potential for seaweed, seaweed by-products and/or seaweed extracts to serve as nutritional supplements in ruminant diets. Several studies have found that inclusion of *Ascophyllum nodosum* in the diet of cattle and sheep can reduce faecal shedding of enterotoxogenic *Escherichia coli* [73]. Addition of *A. nodosum* PTs at 500 µg/mL resulted in significant decreases in *Fibrobacter succinogenes* but not *Ruminococcus albus* or *Ruminococcus flavifaciens* and increased the abundance of *Selenomonas ruminantium*, *Streptococcus bovis, Ruminobacter amylophilus*, and *Prevotella bryantii* [74]. These results are consistent with the observed decrease in fibre degradability observed in vitro in the presence of *A. nodosum* PTs [74]. Orpin et al. [71] also observed increased levels of non-cellulolytic rumen bacteria in the rumen contents of rams consuming a diet containing seaweed. These impacts are not only seen with *A. nodosum*. An in vitro investigation of the impact of *Ecklonia stolonifera* on rumen fermentation observed shifts in the population of cellulolytic rumen microbes [75]. It is possible that the addition of macroalgae alters the availability of polysaccharides required for efficient growth of certain rumen microbes.

To date, few studies have reported the use of genomics and metagenomic (a “multi-omics” approach) techniques to examine the complex impacts that the consumption of macroalgae can have on the composition and function of the microbial community in the gastro-intestinal tract of ruminants. A multi-omics approach has been used to examine the potential value of *A. nodosum* and *Laminaria digitata* as alternative feeds for ruminants and found species-dependent effects on rumen fermentation [58]. In vitro, these species did not significantly alter rumen fermentation patterns, feed degradation or CH_4_ emissions when supplemented at an inclusion level of 5% of dietary dry matter. There were minimal effects on the richness, diversity or composition of rumen bacteria and archaea due to supplementation by either species. Interestingly, the microbial community in the *Laminaria* spp. treatments showed altered carbohydrate metabolism with increased xylan and carboxy-methyl-cellulose digestion compared to control diets [58]. This result is similar to that of Roque et al. [14] and reveals that despite only small shifts in the composition of the microbial community, macroalgae can have significant impacts on the function of the rumen microbiome.

Metagenomics has also been applied to examine the impact of *A. nodosum* in vivo [76]. Inclusion of *A. nodosum* in the diets of rams resulted in a decrease in the total amount of bacteria and archaea in the rumen and an increase in the amount of protozoa. Although only small changes in relative abundance of individual taxa were observed, principal component analysis showed statistically significant clustering of rumen samples based on *A. nodosum* inclusion level [76]. This result indicates that the composition of the rumen microbial community was significantly altered by the inclusion of *A. nodosum* and that this effect was concentration dependent. In addition to changes to the rumen microbial community, Zhou et al. [76] also observed decreased faecal shedding of several serotypes of shiga toxin producing *E. coli*. This suggests that supplementation of diets with seaweed can alter the microbial community in both the upper and lower gastrointestinal tract.

## 4. Seaweed Bioactives and CH_4_ Emissions Reductions

The potential use of phytogenic feed additives has happened largely due to the 2006 EU ban on the use of antimicrobial substances as growth stimulators. However, they are also used as sensory enhancers, technological additives or substances positively affecting the quality of animal products. Halogen compounds including bromoform reduce CH_4_ emissions and react with vitamin B12, thus inhibiting the ability of enzymes from methanogens to produce CH_4_. However, other seaweed bioactives may also impact methanogens. For example saponin compounds are defaunation agents for protozoa and therefore can reduce the population of protozoa in rumen fluid and decrease the methanogens associated with protozoa, and therefore decrease CH_4_ production [77,78]. In addition, tannins are compounds that bind to proteins. Tannin–protein complexes decrease ruminal protein digestibility, but do not increase total tract feed digestibility [79]. Furthermore, it has been shown that antimicrobial peptides (AMPs) can inhibit the methanoarchaea *M. stadtmanae* and *M. smithii* in the human gut microbiome and methanoarchaea are prone to the lytic effects of antimicrobial peptides (AMPs) [13,80,81]. Bioactive compounds in seaweeds other than bromoform could play a role in CH_4_ mitigation in ruminants. Understanding the degree to which structural features in a compound may affect the biological activity of an extract is essential.

### 4.1. Biogenic Halocarbons and Bromoform

Marine algae are an important source of biogenic halocarbons and contribute approximately 70% of the worlds’ bromoform [82]. They produce halocarbons as a defense mechanism against physical and chemical stressors and halocarbons have antibacterial and anti-herbivory functions which protect seaweeds [83]. Red seaweeds contain over 1500 secondary metabolites and along with lichen and fungi can be rich in the halogenated aliphatic organobromine compounds bromomethane and bromoform and chlorine [84]. They have one or two carbon atoms and can lower ruminal CH_4_ production by blocking the function of corrinoid enzymes and inhibiting cobamide-dependent methyl group transfer in methanogenesis. Additionally they can act as terminal electron acceptors. However, emissions of halocarbons create a pool of atmospheric halogen radicals, which directly or indirectly contribute to climate change but most halocarbons produced by seaweeds are short-lived in the atmosphere [85]. A bromochloromethane (BCM) formulation, known to inhibit methanogenesis, was included in the diet of Brahman (*Bos indicus*) [86]. Results found that the BCM formulation fed twice daily to steers at a rate of 0.30 g/100 kg liveweight (LW) with a grain-based feedlot diet, resulted in a significant reduction in CH_4_ production (L/h) equivalent to 93.7% after 28 days of treatment. In addition, 40% of this response could be maintained over a prolonged period (60–90 days), without affecting LW gain and feed intake [87]. However, the authors acknowledged that this formulation was unlikely to be used in agriculture in Australia to reduce CH_4_ emissions due to the fact that un-complexed BCM has an ozone-depleting effect and the manufacture, import and export of ozone-depleting substances, such as BCM, are prohibited by the Australian Government under the Ozone Protection and Synthetic Greenhouse Gas Management Act [85,88]. Furthermore, increased seaweed farming in the tropics (especially Indonesia where seaweed farming has increased 33% in the last decade) has resulted in a decrease in the pH of the ocean and this in turn has an impact on emissions of halocarbons and stratospheric chemistry and the ozone layer [85,89]. Despite this, red seaweeds, in particular *A. taxiformis*, have been researched in recent times for their potential to reduce CH_4_ by incorporation into animal feeds or as supplements for ruminants [11,14,15,30,46]. *A. taxiformis* concentrates halogenated compounds that are known to inhibit cobamide-dependent methanogenesis. In addition to CH_4_ reduction the Bonnemaisoniaceae family (Rhodophyta) are amongst those with the highest and broadest spectrum of antimicrobial activity due to the content and diversity of volatile halogenated compounds [87]. The genus *Asparagopsis Montagne* alone is a particularly prolific source releasing over 100 of such volatile halogenated compounds [88,89]. *A. taxiformis* contains between 0.19 and 4.39 mg g^−1^ DW bromoform. Carpenter and Liss [82] assessed the quantities of bromoform and other bromoalkanes released from brown, red and green seaweeds with a view to determining the impact of bromoform release from seaweeds on the ozone layer. The levels found for each seaweed studied are shown in Table 3. Most species assessed are found in the Northern Atlantic Ocean. Mata [89] also found that the amount of bromoform present in the tissue of *A.taxiformis* was influenced by the availability of total ammonia nitrogen (TAN) and carbon dioxide and the addition of hydrogen peroxide to the growth medium decreased haloperoxidase activity and decreased bromoform levels.

### 4.2. Peptides and Bacteriocins

Peptides and bacteriocins isolated or generated from seaweeds could play a role in suppressing archaea and protozoa involved in the production of CH_4_. Several bioactive peptides have been identified or derived from seaweeds to date and BIOPEP-UWM (http://www.uwm.edu.pl/biochemia/index.php/en/biopep) is a useful resource where many of these peptides are logged. *C. crispus* (Irish moss) is a red seaweed widespread in the Northern Atlantic, which can be rich in protein depending on the season of harvest [90]. There are only a few studies investigating the effect of bacteriocins on CH_4_ emission but Bovicin HC5 and Nisin, a bacteriocin produced by *Lactobacillus lactis* subsp. *lactis*, were shown to decrease CH_4_ emissions by 40% previously. Peptides and bacteriocins have potential to modulate rumen fermentation leading towards increased propionate, thereby decreasing CH_4_ production [20].

Bacteriocins are produced by bacteria and are small, heat-stable peptides that are active against other microbes to which the producer has a specific immunity mechanism. Bacteriocins from lactic acid bacteria and Bacillus species are the most studied. Coincidentally, bacteria from the genus Bacillus are associated in particular with brown algae surfaces and are known to produce the lantibiotic—lichenicidin [91] and all marine seaweeds or invertebrates are suitable habitats for their production and several have already been identified from seaweed-associated bacteria [92,93,94,95,96]. For example, marine bacteria were isolated from brown algae including *Laminaria japonica* [97], *F. serratus* [98] and *F. evanescens* [99] and *U. pinnatifida* [100] and all displayed antimicrobial activities. Suresh et al. [101] identified a bacteriocin produced by *Staphylococcus haemolyticus* methylsulfonylmethane (MSM) isolated from species *U. lactuca, Gracilaria* spp., and *Padina* species. Bacteriocins are thought also to be active against archaea and protozoa [81]. A number of anti-protozoal peptides have also been isolated from marine bacteria associated with seaweeds including Viridamide A isolated from the bacterium *Oscillatoria nigor viridis* which is active against *Leishmania mexicana* and *Trypanosoma cruzi* with an IC_50_ value of 1.1–1.5 µM and Diketopiperazines isolated from marine fungi and active against *Trypanosoma brucei* with IC_50_ values of 40 µM [102].

### 4.3. Phlorotannins

PTs are only found in brown algae, and are polymers of phloroglucinol (1-, 3-, 5-trihydroxybenzene). They are herbivore deterrents and help to protect brown seaweeds from UV radiation [103]. Despite these benefits, tannins are considered anti-nutritional although some authors have reported that it is actually galactolipids that are herbivore deterrents [104]. However, intake of tannins at levels between 2 and 4% were found to have positive effects in ruminants, increasing protein metabolism [105], reducing bloat and acting as anthelmintic agents on gastrointestinal parasites [106,107]. PT concentrations in brown seaweeds are known to be highly variable and likely this is due to variations in biotic and abiotic environmental conditions [108]. Their concentration varies between species, and has been reported to range from 0.5 to 20% dry weight of the alga, although some studies reported concentrations of up to 30% dry weight. The concentration of PTs varies between 0 and 14% of the dry weight of the seaweed. Brown seaweeds from locations at higher latitudes contain more PTs perhaps due to the need of seaweeds to modulate chemical defense production in response to stimuli at these latitudes compared to seaweeds from tropical regions where the environmental conditions are more constant [109]. Several researchers have assessed the quantity of PTs in different seaweed species [110]. Table 4 collates information on the level of PTs found in different brown seaweeds.

Recently, Vissers et al. [111] identified that PTs from *L. digitata* decreased protein degradation and methanogenesis during in vitro ruminal fermentation. PTs were added at a concentration of 40 g/kg and reduced CH_4_ emissions by 40% after 24 h without negatively affecting organic matter digestion or volatile fatty acids. More recently, Moneda et al. studied eight different seaweeds (Brown: *A. esculenta*, *L. digitata*, *P.canaliculata*, *S. latissima*; Red: *M. stellatus*, *P. palmata* and *Porphyra spp*.; Green: *C. rupestris*) that were included in whole oat hay at a rate of 50% and reported no noticeable anti-methanogenic effect [112]. The study reports no correlation between total extractable polyphenols (TEP) and gas production. Only PTs have been reported to have anti-CH_4_ effects, whereas this study did not differentiate between the types of polyphenols nor provide information on how the TEP were quantified, which is known to be inaccurate using the Folin–Ciocalteu method. Furthermore, the study was only performed over 24 h, so it is unlikely that the biopolymers had sufficient time to degrade and release PTs. A study by Pavia and Toth [113] found that nitrogen availability explains some of the natural variation in the PTs content of *F. vesiculosus*, but the light environment has greater importance than nitrogen availability in predicting the PTs content of each species.

### 4.4. Lipids

It is well known that dietary fats/lipids and fatty acids lower CH_4_ emissions from ruminants. The mechanisms responsible for this effect could relate to reduced fermented organic matter in the rumen, effects of fatty acids on rumen methanogens and protozoal numbers as well as unsaturated fatty acids as a hydrogen sink by biohydrogenation [38,114]. The crude fat content of seaweed/algae is classified as at between 1 and 5% of fresh weight [82]. In a recent study by Bikker et al. [115], brown (*L. digitata*, *S. latissima*, *A. nodosum*), red (*P. palmata*, *C. crispus*) and green (*U. lactuca*) seaweed species from different origins in Northern Europe (harvested in Scotland, France or Ireland) reported average crude fat contents of 1.6,1.3 and 2.6% of fresh weight for brown, red and green seaweed, respectively. The lowest crude fat content was observed for *C. crispus* (7 g/kg dry matter (DM)) and the highest content for *A. nodosum* (38 g/kg dry matter (DM)). This study presented the fatty acid pattern of the different algae species and stated that the fatty acid concentration and composition might be relevant for bioactive properties. However, the fatty acid pattern of seaweeds is characterised by a high proportion of polyunsaturated omega-3 and omega-6 fatty acids. In a meta-analysis, Patra [20] found that fats/lipids with high concentrations of C12:0 (lauric acid), C18:3 (linolenic acid) and polyunsaturated fatty acids could be included in the diet up to 6% of the dietary dry matter (DM) to reduce CH_4_ emissions. The analysis of the fatty acid pattern in the study of Bikker et al. [115] showed that lauric acid is of minor importance, whereas α-linolenic acid has a higher proportion in the fatty acid pattern of European seaweeds. In a study by Poulsen et al. [116], it was shown that rapeseed oil supplementation of the diet reduced methanogenic thermoplasmata in lactating dairy cows. The authors discuss that rapeseed oil components maybe responsible for the anti-methanogenic effect and rapeseed oil also contains a high proportion of polyunsaturated fatty acids (PUFAs) similar to seaweed.

### 4.5. Carbohydrates

Seaweeds are rich in polysaccharides, a group of complex carbohydrate polymers, whose level ranges from 4 to 76% of dry weight, with the genera of brown seaweed *Ascophyllum*, red seaweed *Porphyra* and *Palmaria*, and green seaweed *Ulva* having the highest polysaccharide content [60]. Seaweed polysaccharides function in energy storage and to structurally support the cell wall, which typically consists of a fibrillar skeleton and an amorphous embedding matrix [117]. For energy storage, floridean starch (amylopectin-like α-D-glucan) is produced by red seaweeds, whereas laminarin is produced by brown seaweeds [118,119]. Cellulose (1,4-β-D-glucan) is the most common fibrillar skeleton material in seaweeds, and 1,3,-β-D-xylan, 1,4-β-D-xylan and 1,4,-β-D-mannam are also present, depending on algae species and stage of cell wall differentiation [120].

Compared to terrestrial higher plants, seaweed polysaccharides have some similar structural features (4-linked glucan, mannan and xylan) but are unique in the linkage structure of carrageenans, agarans, porphyrans, alginates, fucoidans, ulvans, 1,3-β-D-xylan, mixed linkage xylan, and glucuronans [121,122]. Despite this structural complexity, seaweed polysaccharides with various structural features can be hydrolysed by carbohydrate-active enzymes (CAZyme) encoded within the genomes of gut microbiota (e.g., Bacteroides) during anaerobic fermentation [72]. The rumen microbiota can also adapt to metabolise seaweed polysaccharides. Recently, there has been increasing interest in the use of seaweeds for livestock feed, and the restricted bioavailability of structurally complex seaweed polysaccharides can result in low CH_4_ production [122].

### 4.6. Alkaloids and Saponins

Alkaloids are nitrogenous compounds with antibacterial, antifungal as well as anti-inflammatory activities [123,124] and there is increased interest in bioprospecting seaweed alkaloids for their anti-inflammatory potential. Algae extracts of the genus *Caulerpa* are rich in caulerpin, an indolic alkaloid with proven anti-inflammatory activity. Caulerpin has been described in different species of the genus *Caulerpa*, such as *C. peltata*, *C.racemosa*, *C.cupressoides*, *C. paspaloides*, *C. prolifera*, *C. sertularioides*, *C. mexicana*, *and C. lentillifera*, besides being found in the red algae *Chondria armata.* The analysis of *C. peltata* and *C. racemosa* ethanolic extracts showed caulerpin as one of the main products [125]. Other indolic alkaloids of the genus *Caulerpa* and already identified are racemosin A and B [126], and C [127], and caulersin [128]. Red algae of the genus *Gracilaria* have also been described as important sources of alkaloids [79].

Saponins are phytochemicals which can be found in most vegetables, beans and herbs. The best known sources of saponins are peas, soybeans, and some herbs with names indicating foaming properties such as soapwort, soaproot, soapbark and soapberry [129]. The effects of saponins on rumen fermentation, rumen microbial populations, and ruminant productivity have been reviewed. Saponins from the desert plants *Quillaja saponaria* and *Yucca* have been examined for their CH_4_-reducing potential when fed to dairy cattle [130]. *Quillaja* saponin at 1.2 g/L, but not at 0.6 g/L, lowered CH_4_ production in vitro and the abundance of methanogens (by 0.2–0.3 log) and altered their composition. Ivy fruit saponin decreased CH_4_ production by 40%, modified the structure of the methanogen community, and decreased its diversity. Saponins from *Saponaria officinalis* decreased CH_4_ and abundance of both methanogens and protozoa in vitro [131]. However, in other in vitro studies, *Quillaja* saponins at 0.6 g/L did not lower CH_4_ production or methanogen abundance, and *Yucca* and *Quillaja* saponins at 0.6 to 1.2 g/L even increased archaeal abundance (by 0.3–0.4 log), despite a decrease in protozoal abundance by *Quillaja* saponin [132]. Tea saponins (30 g/day) also did not lower CH_4_ emission from steers or the abundance of total methanogens but increased the abundance of rumen cluster C (RCC) methanogens and protozoa [133]. Over time, the rumen microbiome can adapt to saponins which may explain variations in results of feeding studies with saponin containing plants to date. Thus, the effects of saponins on methanogenesis and methanogen abundance are highly variable among studies carried out to date.

The Chlorophyta are reported to be a rich source of marine saponins as reviewed recently by Feroz [134]. Table 5 shows the saponin content of different seaweeds. The Yucca plant was reported to reduce CH_4_ and methanogens and contains approximately 10% saponins based on the dry weight of the plant. The content of saponins in several seaweeds was reported as being between 13 and 17% of the dry weight of the seaweed [135,136].

## 5. Effect of Seaweeds/Seaweed-Derived Bioactives on the Microbiome of Ruminant Livestock

Novoa-Garrido et al. [137] investigated the effect of a commercial seaweed product with extracts collected from wild seaweeds (*Ascophyllum nodosum*) in Northern Norway on cultivable commensal intestinal bacterial groups. The regular consumption of seaweed polysaccharides is reported to act as a prebiotic promoting the intestinal health and stimulating the growth of beneficial probiotic bacteria such as Bifidobacteria and Lactobacilli [138]. In contrast, the lambs born from ewes in the seaweed-fed group had high mortality. This high mortality in the lambs in the seaweed-fed group is quite remarkable and is explained by inadequate levels of absorbed antibodies caused by mechanisms not related to modulation in the gut microbiota adaptations [139].

The brown seaweed *Saccharina latissima* and the red *Porphyra* spp. were used to evaluate their in vivo digestibility, rumen fermentation and blood amino acid concentrations [62]. The results showed that protein digestibility of the diets including *S. latissimi* were lower than those of *Porphyra* spp. The protein digestibility of *Porphyra*, both in vitro [66] and in vivo [44], was comparable to that of soybean meal. Lind et al. [44] showed that there was no difference in the growth rate of lambs when fed either soybean meal or *Porphyra* spp. as the protein source. The amino acid profile of *Porphyra* sp. was considered as a relevant protein source for ruminants by Gaillard et al. [140].

As mentioned, inclusion of *A. taxiformis* and *A. aramata* in diets of ruminants has shown considerable decreases in enteric CH_4_. Li et al. [141] investigated feeding *A. taxiformis* in diets of sheep and CH_4_ production and found an 80% reduction in CH_4_ compared to a control diet. Similarly, Roque and colleagues [14] found a 67% reduction of CH_4_ from cows fed *A. taxiformis*, while Kinley et al. [15] found reductions of between 40 and 98% CH_4_ in Brahman-Angus cross steers. Other seaweed species have yet to be investigated for their CH_4_ mitigating effects in ruminants. Table 6 summarises in vivo studies in livestock fed different seaweeds where CH_4_ reductions were observed previously.

## 6. Potential for Scale-Up Trials and Economic Feasibility

### Cultivation of Seaweed

In order to build a sustainable industry around CH_4_-inhibiting seaweed cultivation, it is essential to meet the high demand for biomass required as well as to address the environmental constraints regarding sustainability issues and socio-economic aspects. Table 7 details several companies currently active in the area of seaweed cultivation for CH_4_ emissions reductions. Several of these companies have a focus on cultivation of either *A. taxiformis* or *A. aramata.*

## 7. Gaps in Current Knowledge

Several gaps exist in current knowledge regarding the usefulness of seaweeds to tackle climate change through its potential to reduce CH_4_ emissions as a diet supplement or feed for livestock. The potential negative and positive environmental and subsequent economic impacts of seaweed farming at a large scale still remain unanswered. There is a lack of understanding of the requirements and needs for new markets in order to decide which seaweed species to cultivate, what traits to focus on regarding selective breeding and how to reduce the negative compounds in seaweeds, in addition to knowledge concerning optimum cultivation sites and seaweed growth, harvest and processing as well as storage, transportation, cost and sale prices. There is also a lack of evidence that using seaweed-based ingredients to reduce CH_4_ emissions will be competitive, feasible and technically and economically successful and bioeconomic modelling-based approaches are required. Despite these challenges, seaweed use for CH_4_ mitigation provides a new market development opportunity for feed manufacturers and somewhat addresses the issue of limited terrestrial resources for the development of animal feeds.

Potential delivery mechanisms of seaweeds/seaweed bioactive compounds with CH_4_-reducing potential to ruminants could include the use of purified seaweed bioactives provided in a capsule or in injectable or bolus forms [143]. Other options include incorporation of processed crude plant material or extract into feed pellets or solutions; application of fresh seaweed material to the paddock/feed lot; incorporation of crude seaweed material or extract into water sources on the farm. The most appropriate delivery method is impossible to predict presently and will require animal feeding trials suitable to the animal production method used per country. However, the best delivery option will depend on the seaweed stability and activity, the seaweed compound stability, potency, bioavailability and safety. Of great importance also is the palatability of the selected seaweeds chosen for animals. Palatability refers to those characteristics of a feed that invoke a sensory response in the animal Seaweed has a highly variable composition, which depends on the species, time of collection, habitat and on external factors including light, water nutrient content and several other factors [143]. Feed preferences in ruminants are generally associated with digestive modifications. Animals use their senses to learn to associate the post-ingestive effects of the feed with its sensory characteristics. Ruminants generally develop preferences for feeds that will provide a high satiety level rapidly. Thus, palatability measured as the sensory response invoked by the feed integrates its nutritive value [144]. Physical characteristics of the feed including particle size, resistance to fracture and dry matter content contribute to the sensory response invoked by the animal [144]. Feeds that are free of mould and that have been fermented longer (silages) are more palatable for dairy cows. Stems are less palatable. Feed preference in ruminants is affected by smell and several other variables. Flavours include natural flavour such as garlic, anise, and black cumin or artificial flavours such as fruit extracts and chemical products including vanillin and sodium glutamate. They are usually added as dry powders and included in the diet at levels ranging from 0.5 to 1.5% of the feed. Vanilla and fenugreek may be some of the preferred scents [145]. Other studies have identified that older ruminants have a preference for citric tastes and aromas [146]. Orange, citrics, anise, fenugreek, coconut, molasses and maple flavours have been used in adult sheep previously [147]. Sheep showed stronger preference for flavoured feeds than goats in a study carried out by Robertson et al. but both sheep and goats showed a similar pattern of preference across the flavours offered. In general, sheep exhibited significant preference for truffle, garlic, onion, apple, caramel, maple and orange relative to the unflavoured feeds, whereas goats showed significant preference for truffle, onion, apple and garlic [148]. The timing of feeding any new feed ingredient to cows for CH_4_ mitigation is also very important as some ingredients can be detected in the milk or meat, for examples—flavours. Flavours can accumulate in body tissues, especially in fat, and can be transferred to the blood and milk. Flavoured CH_4_ mitigating feeds should therefore be fed after milking and withheld from the cows 4–5 h before milking. Inclusion of seaweeds in the diet of cows could also increases the level of iodine in milk as observed by Antaya who found previously that incremental amounts of *A. nodosum* meal do not improve animal performance but do increase milk iodine output in early lactation dairy cows fed high-forage diets [148].

## 8. Conclusions

The potential of seaweed use to reduce enteric CH_4_ emissions from ruminants depends on a number of factors including the level of the bioactive compound present in the seaweed, which in turn is dependent on seaweed availability and sustainability, harvesting, transport, storage and processing methods employed to formulate seaweed into a feed ingredient. The safety and mechanism of action of the seaweed-derived CH_4_-reducing active ingredient and the palatability of the seaweed/seaweed active ingredient and delivery of these in the diet of the animal are also important considerations. Perhaps the most important consideration is the economic feasibility of seaweed use and processing of seaweeds into extracts for use in animal feed to reduce CH_4_ emissions for ruminants. Several in vitro studies detailing the efficacy of seaweeds in reducing CH_4_ emissions from ruminants have been carried out to date but additional large-scale animal trials are required to determine real efficacy in the field. In addition, details regarding seaweed bioactives and their mechanisms of action will be necessary to ensure seaweed-derived active ingredients for CH_4_ emissions reduction can be sold as animal feed/feed additives in the future.

## Figures and Tables

**Figure 1 animals-10-02432-f001:**
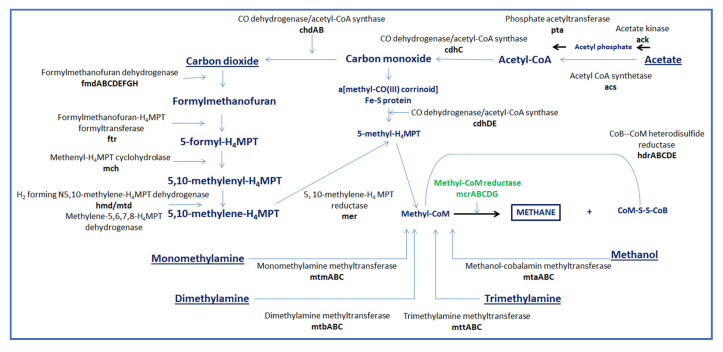
Schematic representation of methanogenesis pathways from carbon dioxide (hydrogenotrophic), acetate (acetoclastic), and mono, di-, tri- methylamine and methanol (methotrophic) pathways.

**Table 1 animals-10-02432-t001:** Summary of seaweeds with CH_4_ mitigation effects in vitro.

Seaweed	Seaweed Dose (% of Dry Matter (DM) or OM Incubated)	CH_4_ Decrease vs. Control	Effects on TGP, TVFA and Digestibility	Reference
*Alaria esculenta* extract (B)	13, 23, 31	Linear ↓ with increasing dose	TGP, n.e.; ↓DOM with ↑ dose	[9]
*Ascophyllum nodosum* (B)	11.1	↓15% at 24 h	↓TGP, ↓TVFA	[10]
*Asparagopsis taxiformis* (R)	5	n.e. at 24 h, ↓74% at 48 h	Not measured	[52]
	1, 2	↓>99%	n.e. on TGP or DOM	[45]
	0.006, 0.013, 0.025, 0.05, and 0.1	↓100 for ≥0.05	↓TGP (24, 48 h) for ≥0.05	[45]
	0.5, 1, 2, 5, 10	↓100% for ≥1%	↓TGP for ≥2%; ↓DOM for 10%; ↓VFA for ≥1%	[11]
	2%	↓100%	↓TGP; ↓TVFA; DOM-72 h, n.e.	[11]
	16.6	↓100%	↓TGP	[49]
	0.07, 0.125, 0.25, 0.5, 1, 2, 5, 10, and 16.8	n.e. for ≤0.5%, ↓85% for 1%, ↓100% for ≥2%	↓TGP for ≥1%; ↓DOM for ≥10%; ↓TVFA for ≥0.5%	[46,47]
	2	↓100%	↓TGP	[53]
*Caulerpa taxifolia* (G)	16.6	↓33%	TGP, n.e.	[49]
*Chaetomorpha linum* (G)	16.6	↓40%	↓TGP	[49]
*Chondrus crispus* (Irish moss) (R)	0.5	↓12%	n.e. on TVFA or DOM	[54]
*Cladophora patentiramea* (G)	16.6	↓66%	↓TGP	[49]
*Colpomenia sinuosa* (B)	16.6	↓49%	↓TGP	[49]
*Cystoseira trinodis* (B)	2, 3.8, 7.4, 13.8	↓73% for ≥3.8% only	↓TGP	[50]
	16.6	↓45%	↓TGP	[49]
*Dictyota bartayresii* (B)	16.6	↓92%	↓TGP	[49]
*Furcellaria* spp. (R)	0.5	↓10%	n.e. on TVFA or DOM	[54]
*Gigartina* spp. (R)	25	↓56%	TGP, n.e.	[51]
*Gracilaria* spp. (R)	2, 4, 5, 7	↓49% for 2%, small ↓ for ≥4%	↓TGP for 2%; other doses, n.e.	[55]
*Gracilaria vermiculophylla* (R)	25	↓41%	TGP, n.e.	[51]
	25	↓37%	↓TGP	[51]
*Hormophysa triquetra* (B)	16.6	↓44%	TGP, n.e.	[49]
*Hypnea pannosa* (R)	16.6	↓43%	TGP, n.e.	[49]
*Laurencia filiformis* (R)	16.6	↓40%	↓TGP	[49]
*Oedogonium* spp. (FW)	16.6	↓30%	↓TGP	[49]
	10, 16.7, 25, 50, 75, and 100	n.e. at ≤25%, ↓17% for 50%, ↓55% for 75%, ↓72.5% for 100%	↓TGP for ≥10%; ↓DOM and ↓TVFA for ≥16.7%	[47]
*Padina australis* (B)	16.6	↓51%	↓TGP	[49]
*Sargassum flavicans* (B)	16.6	↓34%	TGP, n.e.	[49]
*Ulva ohnoi* (G)	16.6	↓45%	↓TGP	[49]
*Ulva* spp. (G)	16.6	↓50%	↓TGP	[49]
	25	↓45%	TGP, n.e.	[51]
*Zonaria farlowii* (B)	5	↓11% at 24 h only	Not measured	[52]

B, brown; DM, dry matter; DOM, degradability of organic matter; F:C, forage:concentrate ratio; FW, fresh water algae; G, green; n.e., no effect; OM, organic matter; R, red; TGP, total gas production; TVFA, total volatile fatty acids.

**Table 2 animals-10-02432-t002:** Chemical composition and macro minerals, trace elements and potentially toxic elements of brown seaweeds.

	Chemical Composition					Macrominerals (g/kg DM)			Trace Elements (mg/kg DM)				Potentially Toxic Trace Elements (mg/kg DM)		
	Ash (% DM)	CP (% DM)	NDF (% DM)	EE (% DM)	DMD (%)	Ca	P	Mg	Fe	I	Zn	Br	As	Pb	Hg
*Saccharina latissimi*	40 *	12–(17.5 *)	40		60	10	3	5	30	960–120	40	550	26–70	0	0.1
*Fucus serratus/spp.*	30	6	26		15–50	9–13	2	7–9	40–310	300	40–50	420	25–40	1	0.1
*Laminiaria digitata/spp.*	25–38	9–12	17–22	0.30	70–75	10–13	3	6	30–179	880	18–25	280	50–70	0	0
*Pelvetia canaliculata*	21	7		2.80		9	1	8	200	250	70	520	50	0	0.1
*Ascophyllum nodosum*	220	4.5–6			32	10–30	1–2	5–10	35–100	<1000	35–100		<3–22		

Adapted from: [61,62,63,64,65,66], [10], [43]. * Protein-enriched *S. latissimi*.

**Table 3 animals-10-02432-t003:** Bromoform levels released from brown, red and green seaweeds per hour (adapted from Carpenter and Liss [82].

Seaweed Species	Bromoform (CHBr_3_) ng g^−1^ h Fresh Weight
Brown	
*Fucus vesiculosus*	4.9
*Fucus serratus*	2.1
*Ascophyllum nodosum*	2.7
*Laminaria digitata*	49.7
*Laminaria saccharina*	32
*Macrocystis pyrifera*	125
Red	
*Meristiella gelidium*	25
*Rhodymenia californica*	47
*Pterocladia capillacea*	500
*Cordllina officinalis*	1.4–20
*Gigartina stellata*	4.1–26
*Asparagopsis spp.*	43–1256
*Chondrus crispus*	0–1.3
*Polysiphonia lanosa*	2.1
Green	
*Ulva intestinalis*	87–192
*Ulva linza*	11
*Ulva spp.*	150
*Ulva spp. (formerly lactuca)*	13.0–150
*Cladophoria albida*	0

**Table 4 animals-10-02432-t004:** Quantity of phlorotannins found in different brown macroalgae (adapted from Lopes et al. [110] and Pavia and Toth [113]).

Brown Seaweed	Phlorotannin Content (mg/Kg)
*Ascophyllum nodosum*	34.9
*Fucus vesiculosus*	42.3
*Cladostephus spongiosus*	81.64
*Cytoseira nodicaulis*	516.24
*Cytoseira tamariscifolia*	815.82
*Cytoseira usnevides*	288.2
*Fucus spiralis*	968.57
*Halopteris filicina*	101.97
*Saccorhiza polyschides*	36.68
*Sargassum vulgare*	74.96

**Table 5 animals-10-02432-t005:** Saponin content of different seaweed species.

Seaweed Species	Saponin Content (% of Dry Weight of Alga)
*Gracilaria crassa*	15
*Gracilaria edulis*	17
*Cymodoceae rotudata*	13
*Cymodoceae serrulata*	14
*Ulva lactuca*	14
*Ulva reticulate*	16
*Gracilaria foliifera*	14
*Kappaphycus alvarezii*	14
*Galidiella accrosa*	13

**Table 6 animals-10-02432-t006:** Summary of seaweeds with CH_4_ mitigation effects in vivo.

Seaweed	Seaweed Dose (% of Dry Matter (DM)) and Animal Trial Used	CH_4_ Decrease vs. Control	Effects of TGP, TVFA and Digestibility	Reference
*Asparagopsis taxiformis (R)*	<2% OM	*Asparagopsis* inclusion resulted in a consistent and dose-dependent reduction in enteric CH_4_ production in sheep over time, with up to 80% CH_4_ mitigation at the 3% offered rate compared with the group fed no *Asparagopsis* spp. (*p* < 0.05).	Sheep fed *Asparagopsis* had a significantly lower concentration of total volatile fatty acids and acetate, but higher propionate concentration. No changes in live weight gain were identified. Supplementing *Asparagopsis* in a high-fibre diet (<2% OM) results in significant and persistent decreases in enteric methanogenesis over a 72 d period in sheep.	[141]
	0.05–0.20% feed	*Asparagopsis* spp. was included in the feed of Brahman-Angus cross steers at 0.00%, 0.05%, 0.10%, and 0.20% of feed organic matter. Emissions were monitored in respiration chambers fortnightly over 90 d of treatment. *Asparagopsis* demonstrated decreased CH_4_ up to 40% and 98%.	*Asparagopsis* resulted in weight gain improvements of 53% and 42%, respectively.	[15]
	5% inclusion rate in OM	Feeding cows a 1–2% dry mass supplement of *Asparagopsis* per day reduced CH_4_ release by up to 95% without altering fermentation processes.	No negative impact on milk taste. No obvious negative impacts on volatile fatty acid production.	[14]
*Ascophyllum nodosum (B)*	2% of diet dry matter (DM)	Limited effect on rumen microbiome.	Feeding 8 Canadian rams with sun-dried seaweed extract (Tasco; Acadian Seaplants Ltd., Dartmouth, NS, Canada) containing a mixture of polysaccharides and oligosaccharides and derived from *Ascophyllum nodosum* resulted in decreased faecal shedding of *E. coli.*	[76]
Seaweed-derived beta-glucans and marine omega-3 oils (fish oil)	5 L (120 g/L) per day of milk replacer (MR) and one of the following: (1) 40 g n-3 PUFA per day; (2) 1 g β-glucans per day (GL) and (3) 40 g n-3 PUFA per day and 1 g/d β-glucans in a 2 × 2 factorial design.	Improved immunity.	44 Holstein Friesian bull calves pre- and post-weaning.	[142]

B, brown; DM, dry matter; OM, organic matter; R, red; TGP, total gas production; TVFA, total volatile fatty acids.

**Table 7 animals-10-02432-t007:** Seaweed cultivation companies with a focus on CH_4_ emissions reduction.

Company	Location	Activities
Symbrosia	USA	Production techniques for *Asparagopsis* spp. seaweeds
Volta Greentech	Sweden	Production of *A.s taxiformis* species
Taighde Mara Teo	Ireland	Production of *A. aramata*
DúlaBio	Ireland	Production of seaweed “blends” for CH_4_ reduction
BMRS	Ireland	Production of *Asparagopsis* spp.
CH_4_ Global	Australia	Global supply of *Asparagopsis* spp.
SeaExpert—Consultoria na Área das Pescas, Lda.	Portugal	Sustainable harvest and supply of *Asparagopsis* spp.
Acadian SeaPlants Ltd.	Canada; Ireland and UK	Sustainable harvest and supply of a range of different seaweeds
SeaLac Ltd.	Ireland	Supply of sustainable seaweeds

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
