# Peer review of "Seaweed and Seaweed Bioactives for Mitigation of Enteric Methane: Challenges and Opportunities"

_animals, 2020, doi:10.3390/ani10122432_

Round 1

Reviewer 1 Report

Dear Authors I recoomed that your think about the target and cut down much of the information that does not fit with that target.  For example the detailed extraction techniques for the classes of compounds.  This Journal is called Animal.  It is not a chemical or medical journal.

However the subject is of great interest and the review is of great importance.

Referees comments on Paper 983302 Entitled Seaweed bioactive ingredients for mitigation of 3 enteric methane: Challenges and opportunities

The paper reviews current literature on methodologies to reduce ruminant methane production.  It then attempts to look at each component of seaweed and how the feeding of seaweed either whole as part of a mixed ration or with specific bioactive ingredients purified from seaweeds may impact on ruminant methane emissions.  Whilst this is a very important and interesting subject I feel the authors need to focus on what exactly the review is about.  They have mentioned very many different aspects of the secondary compounds that are present in seaweeds and the activity that these compounds have been used for especially in the medical profession.  This in my opinion is very interesting but does not fit the title.  They have also gone into great detail of the extraction process for each group of compounds again in my opinion this is not appropriate for this paper in this journal.

I will provide one example here section 4.2 peptides and bacteriocins. Relevance to the manuscript. L 391-394 ‘Atlantic, which can be rich in protein depending on the season of harvest and may be used in human nutrition as a supplement to manage symptoms such as fatigue, asthenia, and weakness and for supporting skeletal muscles in sports athletes [93].  These sentences occur in many parts of the paper and should in my opinion be removed to provide the paper with much needed focus on the actual title.

At other times the authors review the microbiology of sea, stating that sea water contains 1 x 106 bacteria/ml.  The rumen contains 1 x 1012 bacteria/ml.  So it would take 1 x 106 litres of sea water to have the same bacterial load as the 1 ml of rumen fluid!  The microbiology of the seaweed is interesting, and if those microbes were found to have beneficial methane reducing properties they could be fed in the pure form to ruminants at a much lower financial and economic cost than a seaweed supply chain.  Again focus is required with in the context of this paper.

Specific comments

L195 what do you mean by higher dose rates, please specify

L200-205 suggest adding in a sentence on measuring methane output in terms of Kg product produced.  The authors elude to this in terms of reduced animal performance but this approach would highlight the precise issue of reducing methane production whilst maintain or improving animal production.

L 333-337 The authors claim the positive effects of both saponins and tannins on rumen fermentation and use of protein etc.  However, in the interest of balance they should also include the potential negative effects of too much of these compounds on rumen fermentation and protein degradation etc.  Noted this is done later in the paper.

L417 the authors discuss Lactic acid bacteria in their section on bacteriocins.  However no mention is made of the lactic acid bacterial population residing naturally on seaweeds.  There have been recent publications on this within this Journal.  It would be worth mentioning these.

L702-705 This is a contradiction in evidence yet no mention of explanation is made.‘Ewes supplemented with seaweed meal during the indoor winter-feeding period had lower lactic acid bacteriam (spelling) (LAB) counts in their feces compared to ewes receiving a  control diet supplemented with barley. The regular consumption of seaweed polysaccharides is reported to act as a prebiotic promoting the intestinal health and stimulating the growth of beneficial probiotic bacteria such as Bifidobacteria and Lactobacilli [208]’

L 492 Lipids  Again no real evidence and the lipid contents quoted are in the same range as standard terrestrial forages fed.  Authors need to examine data and include on Clover silages and methane emissions.

L775 ‘]. Sheep showed stronger preference for flavoured feeds than did goats in a study carried out by X but both sheep’  Who is X?

Typographical Comments

L40 A. taxiformis as this is the first mention of this species/genus please put name in full.  It is likely that this review will be read by many non-seaweed experts so a little more information is important.

L73 ‘and’ should not be italicized.

L80 ‘small quantities this’ should be ‘small quantities of this’

L106-107 ‘Methanobrevibacter gottcshalkii’  Spelling should be Methanobrevibacter gottschalkii

L 126 ‘synthesis’ should be synthesise

L158 (https://pubmed.ncbi.nlm.nih.gov/1946759)  Please check the link I think it is wrong.

L214-225 why is this section italicised.  Maybe you feel it is a tangent to the main argument.  If so I personally am undecided whether or not it is a necessary diversion.

L 397 ‘ 58 mµmol/g fresh weight’  Are these the correct units?

L762 ‘postingestive’  hyphenate?

L774 Reference 217 not in reference list.

Author Response

Dear reviewer,

We have addressed all of your queries and have amended the revised manuscript accordingly. Please find attached the detailed response to your comments. Kind regards,

Maria

Reviewer 2 Report

Thank you for giving me the opportunity to review this article. Authors have constructed a very interesting review about the use of seaweed as supplement for ruminants to mitigate methane emissions. Below some comments to take into consideration.

L32 Missing space after "currently."

L40 please, italicize A. taxiformis

L57 "bacteriophages"

L60 change effects to affects

L73 and should not be italicized

L81 change were to was

L94 and bacteriophages?

L120-122 names of species should be italicized

L135 names of species should be italicized

L143 change abundances to abundance

L145 do you mean increasing the RO clade abundance related to the SGMT clade?

L158 is this a reference? please, use the format of the journal

L158 change effect to affect

L174 that allows the use of a large

L187 please, delete Norwegian; authors belong to institutions from 3 different countries and the in vitro trial was conducted in Spain

L207 please, delete the definitions of CSM, F and TMR as they are not used in the Table

L214-225 please, check the use of italics

L215 there is a missing space in A.esculenta. Same in L219 and 220

L220 change effects to affects

L230 please, check the use of italics along the table. Units are missing in the table

L238 genus should be italicized (please, check throughout the paper)

L253 change Carrageenan to carrageenan

L281 change Archaea to archaea. same in L 286

L294-295 names of the species should be spelled completely

L320 change principle to principal

L395 please, change Gigartinine to gigartinine

L421-413 please, do not capitalize Marine or Sponge

L432 do not italicize archaea

L475 please, delete the year from the reference

L478 change effecting to affecting

L479 number of the reference is missing

L490 change Lopez to Lopes

Table 6 please, check the spelling of xylomannan

L688 space missing after pH

L713 change were to was

L719 investigated what?

L719-723 it would be useful to mention also if digestibility of diet or performance of animals was decreased

L725 please, recheck as come abbreviations are not used in the table

L728 this line could be removed

Table 10 please, be consistent and use either location and country or just the country

L775 a reference is missing

Author Response

Dear reviewer,

Please find attached our detailed response to your comments highlighted in green. Thank you,

Kind regards,

Maria

Reviewer 3 Report

I suggest a little improvement in tables 2 and 8 in relation to the presentation.

Author Response

We have improved table 2 as far as possible and we have removed table 8 in light of comments from reviewer who requested a reduction in the section concerning characterisation to give a clear focus to the paper.

Round 2

Reviewer 1 Report

Dear Authors,  

My main concern was the length of the submission and the fact that a number of aspects were off focus for both the title and the journal.   I asked you to focus I feel you have not taken my comments on this matter seriously.  I submitted my assessment of your review and les than 36 h later it is back in my inbox.  

The section on chemical extraction which you say you have reduced is still 91% of what it was in the original manuscript.  3 of the 6 sections have not been reduced at all, the other 3 sections have been reduced by a mere 4 lines and in the version sent to me it is certainly not less than 2 pages.  Either the wrong version has been resubmitted or you have ignored my request.  I hope the former not the latter.  I made the strong suggestion to reduce this section because it was in my view a significant distraction from the point of the review and that all the information supplied could easily be done in one simple table well referenced to the original method.  I realise you want to get this published with minimal extra work, but it would be a much better read if the superfluous information was removed.

There is also a typographical mistake in one of your additions on L 513

Author Response

Please see the attachments to view the extensive reduction of version 3 of this paper.

Reviewer 2 Report

Please find attached some extra comments on the revised version of the manuscript. Some of them are the same I made the first time. It seems that the verison of the manuscript authors attached was not the final version as not all the suggestions were replied. Same in the "authors's reply" document, some of the comments are not answered.

Also, it woul be of great help if authors highlight all the changes in the document and refer to these changes in their reply, including line numbers in the new version of the manuscript.

L280 Please, here and in all the paper italicize A. taxiformis

L59 Again, please, change “effect” and “effects” to “affect” and “affects”

L106 and shuld not be in italics

L150 Please, use “et al.” in the refernce format and include the number of reference

L156 Again, change “effect” to affect”. You state that you did but it is not changed in the revised version of the manuscript

L118-129 Please, check the use of italics

L241 Genera names should be italicized. Please, check carefully throughout the paper

L295-297 Names of genera should be spelled completely

L712-715 Was the performance of this animals with reduced methane emissions affected?

L719 check the abbreviations. Some of them are not used in the table

L728 change Hawaii to USA and Scotland to United Kingdom

L770 change “and collegues” to “et al.”

Author Response

December 6th 2020

Response to reviewer 2 comments

L280 Please, here and in all the paper italicize A. taxiformis

Response: We have checked the revised manuscript and we have italicized A. taxiformis throughout.

L59 Again, please, change “effect” and “effects” to “affect” and “affects”

Response: We have checked the revised manuscript and we have, on line L 59 changed effect and effects to affect and affects throughout.

L106 and shuld not be in italics

Response: On L106, we have removed “and” from italics.

L150 Please, use “et al.” in the refernce format and include the number of reference

Response: On line L150 we have used “et al.” as requested by the reviewer and we have included the number of the reference.

L156 Again, change “effect” to affect”. You state that you did but it is not changed in the revised version of the manuscript

Response: We have changed effect to affect and have checked it this time

L118-129 Please, check the use of italics

Response: On line L118 to L129 we have italicised the names of bacteria as requested by reviewer 2. The section now reads: “The Methanobrevibacter clades can be segmented into two subgroups, the M. smithii-gottschalkii-millerae-thaurei (SGMT) (M. smithii, M. millerae, M. thaueri and M. gottschalkii) and M. ruminantinum-olleyae (RO) clade (M. ruminantium and M. olleyae) with the SGMT clade capable of synthesising both mcrI and mcrII and the RO subgroup possessing only mcrI [18, 24]. Expression of both mcrI and mcrII is regulated by H2 availability in the rumen, with mcrI and mcrII expression occurring in the presence of low and high concentrations of ruminal H2 [25]. As a result, a greater presence of SGMT methanogens in the rumen could be evidence of increased abundance of rumen microbes that synthesize H2 [21]. Similarly, as mcrI is produced in a low H2 environment, a greater presence of RO methanogens may be suggestive of a rumen microbiome harbouring a lower abundance of H2 producing and/or greater abundance of H2 utilising microbes.

Similarly, the abundance of specific bacteria, particularly those that produce H2 are associated with CH4 production [18]. In sheep, three different bacterial communities (ruminotypes Q, S and H) were identified associated with methane emissions [22].”

L241 Genera names should be italicized. Please, check carefully throughout the paper

Response: We have checked the manuscript and we have ensured that all Genera names are italicized in the revised manuscript as requested by reviewer 2. On L241 Ulva sp. is now italicised.

L295-297 Names of genera should be spelled completely

Response: L295-297 – we have spelled out the genera name completely as requested.

L712-715 Was the performance of this animals with reduced methane emissions affected?

Response:

L719 check the abbreviations. Some of them are not used in the table

Response: We have checked the abbreviations and we have removed those that are not used in the table. We have removed : DOM, degradability of organic matter and TMR, total mixed ration – neither are used in the table.

L728 change Hawaii to USA and Scotland to United Kingdom

Response: We have changed Hawaii to USA and Scotland to the United Kingdom as requested.

L770 change “and collegues” to “et al.”

Response: We have changed “and colleagues” to “et al.” as requested.
